# Psychosocial and cognitive predictors of academic achievement among higher education students in Southern Ethiopia

Chalachew Kassaw[1]*, Valeriia Demareva[2]

1 Chalachew Kassaw: Department of psychiatry, Dilla University, Dilla, Ethiopia, 2 Valeriia Demareva: Department of Cyberpsychology, Lobachevsky University, Nizhny Novgorod, Russia

* 1234berekassa@gmail.com

## Abstract

### Introduction

Academic achievement is influenced by a combination of personal cognitive abilities, psycho-social resources, and educational context. Core cognitive functions such as working memory and inhibitory control support students' ability to retain, process, and apply information, while coping strategies help manage academic stress. However, research integrating both psycho-social and cognitive predictors of academic success in low-resource university settings remains scarce. This study aimed to examine the combined psycho-social and cognitive correlates of academic achievement among higher education students in Southern Ethiopia.

### Methodology

The study was conducted from August to November, 2024. A total of 30 undergraduate students participated in the study. Academic achievement was measured using Grade Point Average (GPA). Psychosocial factors were assessed with the Folkman and Lazarus coping inventory, the Adult ADHD Self-Report Scale (ASRS v1.1), the Bergen Social Media Addiction Scale (BSMAS), and the Dundee Ready Education Environment Measure (DREEM). Cognitive functions were evaluated using a two-back task (working memory) and a go/no-go task (inhibitory control). Data were analyzed using Spearman correlation, Mann–Whitney U test, Kruskal–Wallis test, multiple linear regression, and ANOVA.

### Results

Sex differences were observed, with male students demonstrating higher GPA compared to female students. Faster reaction time in the two-back task was significantly associated with better GPA. Higher use of self-controlling coping strategies was marginally associated with GPA in regression modeling, but ANOVA indicated a

**Data availability statement:** All relevant data are within the manuscript and its supporting information files.

**Funding:** The author(s) received no specific funding for this work.

**Competing interests:** The authors have declared that no competing interests exist.

significant contribution of this coping style to the explained variance in academic performance. Other psycho-social variables, including academic self-perception, social media addiction, and ADHD symptoms, were not significantly related to GPA. Overall, psycho-social factors, particularly coping strategies and gender, were stronger predictors of academic achievement than cognitive measures in this sample.

## Conclusion

The findings highlight the critical role of coping mechanisms and gender differences in academic performance among Ethiopian university students. These results underscore the need for psycho-social support interventions alongside cognitive skills development to enhance academic success in low-resource educational environments.

---

### 1. Introduction

Academic success in higher education is a multifaceted outcome influenced by intellectual, psychological, and social factors. Beyond mere grades, academic achievement holds profound significance for students, shaping their future job prospects and overall well-being [1].

It is a basis for personal and national prosperity but is hampered by the interplay of personal, social, and environmental conditions . The most common and traditional way of measuring students' academic capacity is either based on their previous score or current intelligence quotient which often overlook the interaction between psycho-social and cognitive processes in determining students' and institutions' academic performance in determining students and education institution academic competency.

There is growing scientific evidence that support the role of students psychological and cognitive process on their academic journey and consistency. In low-resource setting, there is limited evidence on exploring how the psycho-social and cognitive elements interact to predict student academic achievement.

In Ethiopia, where the higher education sector has rapidly expanded in recent years [2], student achievement rates remain a concern [3]. However, the academic achievement of students is greatly affected with multifaceted factors such as quality concerns, high attrition, student preparedness, socio-economic issues, cultural transitions, and resource limitations. Identifying what enables some Ethiopian university students to excel while others struggle is crucial for informing interventions in this low-resource context.

As a first step in addressing this issue, a recent systematic review synthesized findings from Ethiopian studies and highlighted numerous psycho-social determinants of academic performance [4]. That review revealed that stress and mental health challenges (e.g., high perceived stress, poor sleep quality, depression, suicidal ideation) and negative self-perceptions (e.g., low self-esteem, poor academic self-concept) were consistently linked to poorer academic outcomes [5,6].

Similarly, mal-adaptive behaviors and social factors – such as excessive social media use, substance use, and financial strain – were associated with lower grade point averages (GPAs) [7–9]. These findings underscored that Ethiopian students' academic achievement is intertwined with their psychological well-being and social environment [10–12], extending global research on academic success into the Ethiopian context.

Building on the review, a second phase of this research program empirically tested key psycho-social predictors in a sample of Ethiopian undergraduates (at Dilla University) [3]. That study confirmed several predictors identified in the review. Notably, students who relied heavily on emotion-focused coping strategies were significantly more likely to have low academic achievement. In particular, high emotion-oriented coping (e.g., feeling overwhelmed by stress) was associated with four-fold higher odds of low GPA, suggesting that struggling to manage academic stress can undermine performance. Likewise, students with poor academic self-perception (low academic self-confidence) had nearly double the odds of low achievement. By contrast, those who limited their social media use were about 68% less likely to be low achievers, aligning with evidence that excessive online engagement detracts from study time and focus. Another important finding was the role of gender: female students outperformed their male peers on average. This countered a common assumption that female students might underachieve [13–16], and instead indicated that in this context female students had equal or better academic outcomes than males [17]. Overall, the second-phase study validated that how students cope with stress and perceive themselves, along with certain behaviors (like media use), significantly relate to their academic success or failure in Southern Ethiopia [3].

While these two studies enriched our understanding of psycho-social influences, a key knowledge gap remained. Cognitive factors – such as students' working memory capacity, attentional control, and symptoms of attention-deficit/hyperactivity disorder (ADHD) – were not examined in prior phases. These cognitive factors are preferred over other domains such as set-shifting, meta-cognition, and executive functions because of their foundational and direct roles in academic success across various educational settings [18]. The executive function and cognitive load theory supports working memory and attentional control are a central component of executive functions vital for problem solving and handling huge informations without difficulty [19]. The role of ADHD symptom on learning difficulty is related with the direct effect on working memory and extraneous stimulus attention distraction [20]. There is robust evidence globally that such cognitive abilities can shape academic performance. For instance, studies suggest that working memory, the ability to hold and manipulate information in mind is a strong predictor of academic achievement in areas like reading and math [21]. Inhibitory control and processing speed, often measured via reaction times on tasks have also been linked to better learning outcomes, especially in childhood and adolescence [21,22]. Furthermore, ADHD-related attentional problems are known to correlate with lower academic performance in college students [23]– particularly symptoms of inattention can undermine study habits and exam performance. However, these cognitive predictors have been largely understudied in Ethiopian higher education. The omission is notable, as cognitive skills might play an important role even when resources and prior educational quality vary. It remains unclear whether, in a low-resource university setting, cognitive factors add additional explanatory power for GPA beyond psycho-social variables.

To address this gap, the present study (the third phase of our research program) integrates cognitive measures into the established psycho-social framework. Based on the earlier findings, we selected the strongest psycho-social predictors of academic achievement (such as coping strategies and gender) and added standardized cognitive assessments of working memory, inhibitory control, and ADHD symptoms. Specifically, we employed a 2-back task to gauge working memory, a Go/No-Go task to evaluate inhibitory control (impulse regulation and attention), and a validated self-report scale for ADHD symptoms. There is a dynamic interaction between psycho-social and cognitive factors that shapes the academic success of students from under-resourced settings. Financial stressors, inadequate study spaces, and limited technological access in such contexts amplify the interplay between cognitive and psycho-social domains in influencing academic performance. For example, students with stronger attentional control may maintain focus despite disruptive living conditions, whereas those with weaker attentional control are more vulnerable to the adverse effects of environmental stressors. Similarly,

students with well-developed working memory can efficiently process and retain information even when learning materials are limited. Previous studies in Ethiopia have largely examined isolated factors of academic achievement, and there is a lack of comprehensive, integrated models assessing the joint predictive power of diverse psycho-social and cognitive variables on academic performance in higher education. Understanding the combined influence of psycho-social and cognitive predictors is not only scientifically intriguing but also practically important: in a resource-limited educational context, it can help target the most impactful factors to improve student outcomes.

The aim of the current study was to examine the associations between psycho-social and cognitive factors and students' GPA among undergraduate students at Dilla University, Southern Ethiopia.

Hypotheses: H1 (Psychosocial predictors). Adaptive coping strategies and positive self-perceptions will be positively associated with higher GPA, whereas mal-adaptive coping strategies (e.g., excessive emotion-focused or distancing coping) and negative self-perceptions will be negatively associated with GPA. In addition, sex is expected to emerge as a significant predictor of academic achievement.

H2 (Cognitive predictors). Stronger working memory and inhibitory control (evidenced by higher task accuracy and faster reaction times) will be positively associated with higher GPA, while elevated ADHD symptoms will be negatively associated with GPA. Understanding students' psycho-social and cognitive strengths and vulnerabilities is essential for developing targeted interventions to enhance learning outcomes and for preparing a competent, skilled workforce.

Through this study, we seek to extend the literature on academic performance predictors by bridging psycho-social and cognitive perspectives, and to provide evidence that can inform interventions to support student success in Ethiopia and similar low-resource educational settings.

## 2. Materials and methods

### 2.1. Study sample and period

**2.1.1. Sample size calculation.** $N = L/f^2 + k + 1$, whereas $N$ = required sample size, $L$ = a value derived from the non-centrality parameter of the F-distribution depending on the desired Power $(1 - \beta)$, Alpha $(\alpha)$, and the degrees of freedom, $f^2$ = Cohen's effect size for multiple regression, $k$ = number of predictors (independent variables), and $+1$ = for the intercept. The sample size was calculated using G*Power version 3.1.9.7 based on statistical power analysis principles with values of ($f^2 = 0.35$, $\alpha = 0.05$, Power = 0.80, Number of predictors = 4). For these parameters, the non-centrality parameter $(L)$ is approximately 8.75, yielding $N \approx 30$.

**2.1.2. Study participants.** The participants were undergraduate students enrolled at Dilla University, College of Health Sciences. The final sample size for the study was 30 students (20 males). The mean (standard deviation) age of participants was $22.3 \pm 4$, ranged 18–25 years old. The participants were from college of health science department of medicine [5], health officer [4], midwifery [4], anesthesia [3], nursing [4], psychiatry [3], laboratory [4] and environmental health science [3]. All batches of the year in each department were included proportionally on the final sample size.

**2.1.3. Sampling technique.** The participants were selected using convenience sampling methods. To recruit volunteers, several strategies were employed, including classroom announcements, distribution of flyers, and online advertisements in settings where the target population was accessible. Classroom announcements outlined the study objectives and potential significance within the existing academic context. Flyers with an attractive design were distributed in person to students on campus, while online advertisements were shared through available platforms to reach students who were not physically present.

**2.1.4. Inclusion and exclusion criteria.** The inclusion criteria were limited to regularly enrolled students in the designated departments who were present and competent to participate at the time of data collection. Students with previous clinically diagnosed ADHD symptoms or mental illness, and acute physical or mental health conditions that hindered their ability to complete the survey and experimental procedures were excluded from the current study.

**2.1.5. Study period.** The study was conducted from August 15, 2024, to November 15, 2024.

## 2.2. Data collection tools

**2.2.1. Cognitive fun application.** Cognitive Fun is an advanced web-based application designed to assess and enhance various cognitive functions. It is user-friendly and accessible to individuals of all ages and abilities. Cognitivefun. net platform is a popular website used for cognitive assessment and brain training using various types of games. The website was developed on 2008 and can be accessed on (https://cognitivefun.net). The application features a simple interface with clear instructions, making it easy to navigate and understand the cognitive tasks. It is specifically designed to measure different cognitive domains through interactive game tasks, providing users with feedback, tracking progress, and helping improve cognitive abilities [24,25].

The psychometric property for the whole application is highly dependent on the specific nature and design of the games. The TRIAL period of each cognitive task is different. Before the actual data collection, the purpose of the study, voluntarism, confidentiality, and right to withdraw at any time was discussed with study participants. The principal investigator, and two research assistants were involved during the procedure of task-based data collection. The procedure for general data collection using this application is software configuration/task programming then exposure to computer interface and record the data. In this study, inhibitory control (Go/No-Go Task) and working memory (N-Back Task) cognitive domains were selected.

**2.2.2. Go/No-Go task.** This task measures response inhibition capacity. Participants were required to respond when they perceived a target stimulus but refrain from responding when a non-target stimulus appeared. To assess inhibitory control, the application presented users with Go/No-Go visual stimuli, such as full green and striped circles interchangeably. The sensitivity and validity of the task vary depending on study design, participant characteristics, and the chosen cutoffs [26]. In individuals with ADHD symptoms and impaired inhibition, the task demonstrates high sensitivity [27]. There is strong construct validity for inhibitory control, with commission errors reflecting the inability to suppress a dominant response [28]. Initially, participants completed about 15–20 practice trials with a Go (80%) to No-Go (20%) ratio in order to learn to respond appropriately to "Go" stimuli and to practice inhibiting responses to "No-Go" stimuli [29]. During the practice period, participants were expected to achieve a score of approximately 70–80% and received immediate feedback to guide their performance in the actual trial. To guide their actual trail, participants completed 30 trials, and their score was included on data analysis. When the full green circle appeared, participants clicked the circle; otherwise, they refrained from pressing the button (**Fig 1**).

**2.2.3. N-Back task.** This task was designed to measure working memory capacity. For this study, the 2-back task was used to assess participants' ability to hold and manipulate stored information. Participants monitored a sequence of stimuli and indicated when the current stimulus matched the one presented two trials earlier. The N-back task is highly sensitive to working memory load, consistently demonstrating decreased accuracy and increased reaction times as the *n* level increases [30]. The specific sensitivity and specificity values of the task vary across different populations. The intraclass correlation coefficients (ICCs) for N-back accuracy and reaction time typically range from approximately 0.60 to 0.90, depending on sample characteristics [31]. The N-back task has shown robust construct validity as a measure of working memory [32]. Participants completed two free practice sessions lasting 15–20 minutes for the 2-back experimental blocks. This initial step was designed to familiarize participants with the rules, and feedback scores were provided during the practice phase. The practice scores were not included in the main study. During the actual task, participants were required to click when a displayed object (e.g., balls, pencils, hats, hearts) matched the one presented two trials earlier. Each participant completed 20 trials, and their scores were used for analysis [33] (**Fig 2**).

**2.2.4. Grade Point Average (GPA).** GPA is a weighted average of grades, reflecting overall academic performance [34]. Within the current study, GPA performed as a dependent numerical variable.

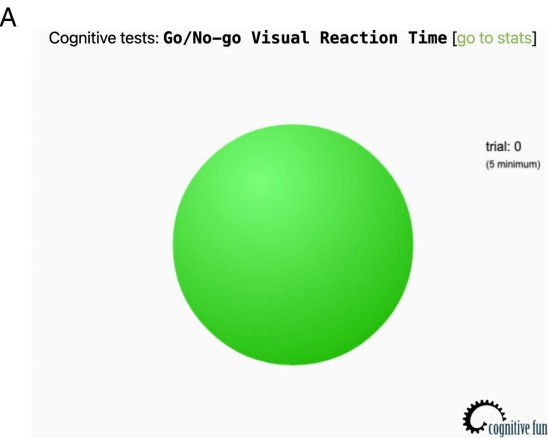

A
Cognitive tests: **Go/No–go Visual Reaction Time** [go to stats]

trial: 0
(5 minimum)

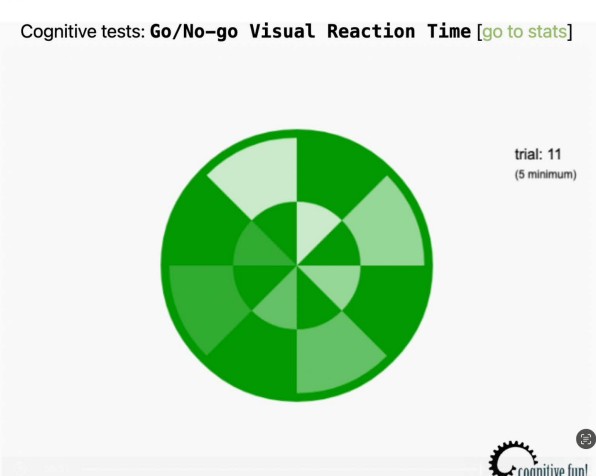

B
Cognitive tests: **Go/No–go Visual Reaction Time** [go to stats]

trial: 11
(5 minimum)

**Fig 1. Stimuli in Go/No-Go Task: (A) – target, (B) – non-target.** The following numerical variables were obtained for 2-Back Task: average (RT average (Go/No-Go)) and standard deviation (RT deviation (Go/No-Go)) of reaction time (ms).

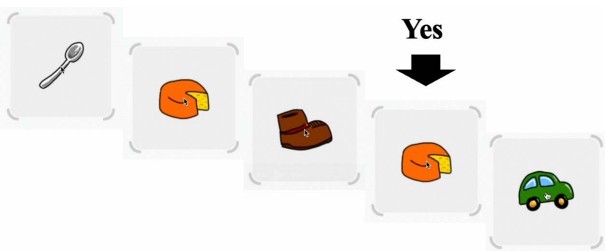

**Fig 2. Stimuli example and target situation ('Yes') for 2-Back Task.** The following metrics were obtained for 2-Back Task: correct response ratio (Correct ratio (2-Back)) and average response time (RT average (2-Back)) (ms).

Psychosocial variables: The following psychological and social measurements were included based on previous literature review and empirical study data and shown as a strong predictor academic achievement in higher education student in low-resource setting. The measurements were also aligned with study objective in assessing the combined effect of psycho-social and cognitive predictors on students' academic success.

**2.2.5. Coping strategies.** Coping strategies were assessed using the coping framework by Folkman and Lazarus [35]. The inventory included 66 items grouped into eight sub-scales:

1. Confrontive Coping – strategies reflecting aggressive efforts to alter the situation;

2. Distancing – cognitive efforts to detach oneself from the situation;

3. Self-Controlling – attempts to regulate one's own feelings and actions;

4. Seeking Social Support – efforts to obtain informational, tangible, or emotional support;

5. Accepting Responsibility – acknowledging one's role in the problem with a concomitant theme of trying to put things right;

6. Escape–Avoidance – wishful thinking and behavioral efforts to escape or avoid the problem;

7. Planful Problem-Solving – deliberate problem-focused efforts to alter the situation;

8. Positive Reappraisal – efforts to create positive meaning by focusing on personal growth.

Items were rated on a Likert-type scale indicating the frequency of use. Subscale scores were calculated by summing responses to the corresponding items and were considered as independent numerical variables.

The internal consistency (Cronbach's alpha) values often range from 0.70–0.90 among higher education student. ((Senol-Durak et al., 2011). The content, criterion, construct and convergent validity of the tool showed acceptable in higher education population. The cronbach's alpha value in this study was 0.81. Students' perceptions of academic environment

The Dundee Ready Education Environment Measure (DREEM) assesses the educational environment in higher education using a 50-item survey. Scores range from 0 (strongly disagree) to 4 (strongly agree), with higher scores indicating a more positive learning environment [36]. A total DREEM score (maximum 200) and scores for each subscale were calculated. The DREEM has been translated and validated in numerous languages and cultures across various countries. Nine items are negatively worded and require reverse scoring. The items are grouped into five sub-scales: Students' Perceptions of Learning (SPL) (12 items), Students' Perceptions of Teachers (SPT) (11 items), Students' Academic Self-Perceptions (SASP) (8 items), Students' Perceptions of Atmosphere (SPOA) (12 items) and Students' Social Self-Perceptions (SSSP) (7 items). The Cronbach's alpha for this measure was 0.84 [37].

**ADHD Symptoms:** The Adult Self-Report Scale (ASRS v1.1), an 18-item questionnaire with a five-point Likert scale (0 = never, 0 = rarely, 1 = sometimes, 1 = often, 1 = very often), was used to assess ADHD symptoms. It was developed by World Health Organization (WHO) and Harvard Medical School, and based on the diagnostic criteria of the DSM-IV-TR. It evaluates nine attention deficit symptoms and nine hyperactivity (motor and verbal) symptoms over the past six months. The tool demonstrated high internal consistency (Cronbach's alpha = 0.88) [38,39]. Symptom severity was measured across three ADHD dimensions: inattention, hyperactivity, and total score. Participants were categorized into groups based on two diagnostic criteria: presence of ADHD symptoms and without ADHD symptom group [40]. The scores for all 18 items are summed to get a total score ranging from 0 to 18. A score > 5 for inattention or hyperactivity was recognized as significant ADHD symptom. The total ADHD score, as well as scores for inattention, hyperactivity (total, motor, and verbal) were considered as independent numerical variables.

**2.2.6. Social media addiction.** The Bergen Social Media Addiction Scale (BSMAS) is a six-item scale used to assess the severity of social media addiction. The tool has acceptable good test-retest reliability. It evaluates difficulties related to

excessive social media use over the past year. The scale has been cross-culturally validated and shows strong sensitivity and specificity for problematic usage. With a Cronbach's alpha of 0.91, it exhibits high internal consistency. Responses were scored on a five-point Likert scale, with total scores ranging up to 30. Higher scores indicate potential problematic usage [41].

**2.2.7. Data collection procedure.** Participants attended a single testing session. After providing informed consent, they completed a series of assessments. First, they filled out an online survey covering socio-demographic, academic, and psycho-social information. Next, they performed cognitive tasks on CognitiveFun.net, with adequate training and practice trials provided for both inhibitory control and working memory tasks. Inhibitory control ability was measured using a simple reaction time task, where participants responded as quickly as possible to a target stimulus. Working memory capacity was assessed using the 2-back task. Participants were presented with a sequence of visual stimuli and asked to identify whether the current stimulus matched the one presented two trials earlier.

**2.2.8. Ethics approval and consent to participate.** The study received ethical approval from the Dilla University College of Medicine and Health Sciences Institutional Review Board (Approval No. duirb/802/2024). The informed written consent was obtained from all participants prior to their enrollment in the study. This process involved providing each participant with a detailed information sheet outlining the study's purpose, procedures, potential risks and benefits, confidentiality measures, and their right to withdraw at any time. Participants were given ample time to review this information and ask questions before signing a consent form, which was then retained securely. Ethical considerations, including confidentiality and data security, were strictly adhered to throughout the study.

**2.2.9. Inclusivity in global research.** Additional information regarding the ethical, cultural, and scientific considerations specific to inclusivity in global research is included in the Supporting Information (S1 File).

## 2.3. Data analysis

Statistical analyses were performed using R Studio (v. 2024.04.2 Build 764). Spearman rank correlation analysis was used to examine associations between GPA and cognitive/psycho-social variables. To assess potential group differences in GPA based on categorical variables (i.e., gender and ADHD status), nonparametric tests were employed: Mann–Whitney U tests for binary groups and a Kruskal–Wallis test for the three-category ADHD grouping. Prior to modeling, continuous predictors were standardized (z-scores). A multiple linear regression model was fitted to identify significant predictors of GPA. The model was constructed based on correlation and comparative screening results, with attention to multicollinearity. To evaluate the contribution of each predictor, an analysis of variance (ANOVA) was performed on the fitted model. All tests were two-tailed, with a significance threshold set at $p < .05$.

## 3. Results

### 3.1. Sociodemographic characteristics

A total of 30 participants with 100% response rate were enrolled in the current study. The mean (standard deviation) grade point average among participants was $3.39 \pm 0.29$ (Table 1)

### 3.2. Correlation analysis between GPA and numerical variables

Spearman rank correlation coefficients were calculated for all cognitive and psycho-social variables included in the study. Significant positive correlations were found between GPA and three coping strategies (Confrontation ($\rho = 0.40$, $p < 0.05$), Self-controlling ($\rho = 0.43$, $p < 0.05$), Distancing ($\rho = 0.44$, $p < 0.05$)), as well as between cognitive performance measures (Correct ratio ($\rho = 0.52$, $p < 0.01$) and RT average ($\rho = -0.56$, $p < 0.01$) in the 2-Back task; RT deviation in Go/No-Go Task ($\rho = -0.48$, $p < 0.01$)). No significant correlations were found between GPA and hyperactivity-related symptoms (inattentiveness, motor hyperactivity, verbal hyperactivity, and total hyperactivity), social media addiction, or academic

**Table 1. The descriptive statistics result of participants attending in Dilla university (n = 30).**

| Variables | All sample (n = 30) | Male (n = 20) | Female (n = 10) |
|---|---|---|---|
| Age (Mean±Standard deviation) | 22.3±4 | 22.1±3.18 | 23±2.12 |
| GPA (Mean±Standard deviation) | 3.39±0.29 | 3.45±0.28 | 3.28±0.29 |
| Psychosocial factors | | | |
| Confrontive coping | 15.13±2.37 | 15.5±2.19 | 14.4±2.19 |
| Distancing | 13.13±2.49 | 12.95±2.78 | 13.5±2.78 |
| Self-controlling | 17.17±2.98 | 16.85±3.3 | 17.8±3.3 |
| Seeking social support | 21.37±3.27 | 21.45±3.58 | 21.2±3.58 |
| Accepting responsibility | 9.6±2.3 | 9.95±2.35 | 8.9±2.35 |
| Escape-Avoidance | 20.97±2.81 | 20.7±2.75 | 21.5±2.75 |
| Planful problem-solving | 19.6±2.94 | 19.45±3.24 | 19.9±3.24 |
| Positive reappraisal | 25.2±3.81 | 25.05±4.17 | 25.5±4.17 |
| ADHD Inattentiveness | 5.53±2.46 | 6.2±2.24 | 4.2±2.24 |
| ADHD Hyperactivity | 5.07±2.73 | 5.7±2.74 | 3.8±2.74 |
| ADHD Total | 10.6±4.57 | 11.9±4.13 | 8±4.13 |
| Social media addiction | 15.47±4.62 | 15.5±4.84 | 15.4±4.84 |
| Academic self perception | 24.2±3.24 | 24.05±3.33 | 24.5±3.33 |

self-perception. In addition, no significant relationships were observed between GPA and most coping strategies, including Escape-Avoidance, Seeking Social Support, Accepting Responsibility, Planful Problem-Solving, and Positive Reappraisal (p > 0.05).

### 3.3. Group differences in GPA by categorical variables

Nonparametric group comparison tests were conducted to evaluate differences in GPA based on categorical factors, including gender and ADHD-related classifications. Mann–Whitney U tests were used for binary groups, and a Kruskal–Wallis test was applied for the three-category ADHD variable (Table 2).

### 3.4. Predictor selection

In the sections above, a total of 18 psycho-social and cognitive variables were screened as potential predictors of academic performance (GPA). As demonstrates in 3.1, six variables showed statistically significant correlations with GPA (p < .05): RT average (2-Back), Correct ratio (2-Back), RT deviation (Go/No-Go), Distancing, Self-controlling, and Confrontive coping.

To reduce redundancy and multicollinearity, only one working memory index was selected. RT average (2-Back) was retained due to its strong negative correlation with GPA ($\rho = -0.56$, p < .01) and its conceptual informativeness as a measure of processing efficiency. Although RT deviation (Go/No-Go) also correlated significantly with GPA ($\rho = -0.48$, p < .01), it was moderately correlated with RT average (2-Back) ($\rho = 0.49$), and therefore excluded from the model.

Among psycho-social variables, both Self-controlling ($\rho = 0.43$, p < .05) and Distancing ($\rho = 0.44$, p < .05) coping strategies were included. These variables were not strongly correlated with each other and reflected distinct self-regulatory tendencies. Confrontive coping, though also significantly associated with GPA, was not selected due to overlapping variance with Self-controlling and a slightly weaker effect size.

In line with findings in 3.2 and to control for potential confounding due to sex-related differences in academic outcomes, Sex was included as a covariate in the final model. The resulting regression model thus included four predictors: Sex, RT average (2-Back) (standardized), Self-controlling (standardized), and Distancing (standardized).

**Table 2. Group differences in GPA by categorical predictors (n = 30).**

| Variable | Test | P |
|---|---|---|
| Sex | Mann–Whitney U | 0.158 |
| ADHD presence | Mann–Whitney U | 0.897 |
| ADHD group | Kruskal–Wallis | 0.253 |

The Mann–Whitney U test comparing GPA between students with and without ADHD symptoms yielded a non-significant result (U = 108.00, p = .897). The Kruskal–Wallis test revealed no statistically significant differences in GPA across ADHD diagnostic groups (No ADHD symptoms, Mixed ADHD symptoms, Hyperactive type), H(2) = 4.08, p = .253. Although the difference in GPA between male and female students did not reach statistical significance (U = 132.50, p = .158), the result indicated a possible trend warranting further analysis.

### 3.5. Multiple regression analysis of GPA predictors

Based on the correlation and comparative analysis results, a multiple regression model was constructed using standardized predictors to identify the best predictors of GPA. The model included Sex, RT average (2-Back), Distancing, and Self-controlling. For the 2-Back task (p = 0.06), there was a trend towards significance for average reaction time (RT) with student grade point average. However, there is a wide confidence interval indicating considerable variability and uncertainty around this estimate.

The regression model was statistically significant (F(4, 25) = 7.65, p < .001) and accounted for approximately 55% of the variance in GPA (adjusted $R^2$ = .478). Regression coefficients, standard errors, t-values, and 95% confidence intervals are presented for each predictor (**Table 3**).

### 3.6. The analysis of variance (ANOVA) of predictor variable

The analysis of variance (ANOVA) for the regression model predicting GPA. The model included four predictors: gender (Sex), standardized RT average (2-Back), standardized Distancing, and standardized Self-controlling. Each predictor's contribution to the explained variance was assessed using F-tests. The ANOVA indicated that the overall regression model was statistically significant. Among the predictors, Sex (F = 5.55, p = .027) and Self-controlling (F = 7.56, p = .011) made significant contributions to the model. RT average (2-Back) showed a marginal effect (F = 3.90, p = .060), while the effect of Distancing did not reach statistical significance (F = 2.59, p = .120). The residual variance remained significant, indicating that a portion of GPA variability was not accounted for by the predictors in the model. The residual term reflects unexplained variance in GPA after accounting for the predictors (Table 4).

### 3.7. Plotted graph for variables included in regression model

To visualize the relationship between the predictors and academic achievement, we plotted the effects of each variable included in the final regression model. The figure presents the predicted GPA values as a function of self-controlling and distancing coping strategies, reaction time in the 2-back task, and gender. Continuous predictors were standardized prior to analysis. Each panel shows the model-predicted GPA trend with a 95% confidence interval, along with individual data points (**Fig 3**).

## 4. Discussion

### 4.1. Key findings

This study examined both psycho-social and cognitive predictors of GPA among Southern Ethiopian university students. Consistent with our hypotheses, we found that coping style and gender were significant predictors of academic

**Table 3. Multiple regression model predicting GPA (standardized predictors).**

| Predictor | B | SE | T | P | 95% CI LL | 95% CI UL |
|---|---|---|---|---|---|---|
| Constant | 3.458 | 0.047 | 73.37 | <.001 | 3.36 | 3.555 |
| Sex | −0.197 | 0.083 | −2.36 | 0.027 | −0.368 | −0.025 |
| RT average (2-Back) | −0.085 | 0.043 | −1.97 | 0.060 | −0.173 | 0.004 |
| Distancing | 0.068 | 0.043 | 1.61 | 0.120 | −0.019 | 0.156 |
| Self-controlling | 0.112 | 0.041 | 2.75 | 0.011 | 0.028 | 0.197 |

Note. All predictors were standardized before entry. B = un-standardized coefficient; SE = standard error; CI = confidence interval; LL = lower level; UL = upper level.

**Table 4. ANOVA table for regression model with standardized predictors.**

| Predictor | SS | Df | F | P |
|---|---|---|---|---|
| Sex | 0.24 | 1.0 | 5.55 | 0.027 |
| RT average (2-Back) | 0.17 | 1.0 | 3.9 | 0.060 |
| Distancing | 0.11 | 1.0 | 2.59 | 0.120 |
| Self-controlling | 0.33 | 1.0 | 7.56 | 0.011 |
| Residual | 1.09 | 25.0 | – | – |

Note. SS = Sum of Squares; df = degrees of freedom; F = F-statistic; p = significance level. Predictors were standardized prior to entry into the model.

achievement. In particular, a higher tendency to use a self-controlling coping strategy was associated with significantly better GPA. Students who reported frequently using self-controlling coping (i.e., deliberately managing their emotions and impulses when facing academic stress) tended to earn higher grades, suggesting that this form of self-regulation benefits academic performance. Additionally, gender emerged as a significant factor on the adjusted regression model, with female students achieving lower GPAs on average than male students (controlling for other variables). This gender effect contradicts the pattern observed in our prior work [3], and indicates that female students in this context are more likely to be low achievers compared to males, as it was obtained in other studies [13–16].

Two other predictors showed noteworthy but non-significant trends. The coping strategy of distancing (emotionally disengaging or minimizing the importance of stressors) exhibited a marginal association with GPA. Specifically, there was a trend for higher use of distancing coping to be linked to lower academic achievement (p approaching significance). In other words, students who often "distanced" themselves from academic stressors tended to have slightly lower GPAs, though this pattern was not statistically robust. Meanwhile, performance on cognitive tasks – particularly reaction time (RT) on the Go/No-Go task – also showed a trend in the expected direction. Faster reaction times (indicating quicker information processing and perhaps better attentional control) were modestly associated with higher GPA, but this effect only approached significance. By contrast, other cognitive measures (such as self-reported ADHD symptom levels) did not show strong relationships with GPA in our sample. Taken together, these results suggest that psycho-social factors, especially coping strategies, had more pronounced effects on academic performance than the cognitive factors measured, although cognitive processing speed did hint at a possible influence.

### 4.2. Comparison with previous literature

Our findings both align with and extend prior research on academic achievement predictors. The importance of coping strategies resonates with earlier studies in Ethiopia and beyond. In our previous phase, mal-adaptive emotion-focused coping was a clear risk factor for poor grades [3]. The current result that self-controlling coping correlates with higher

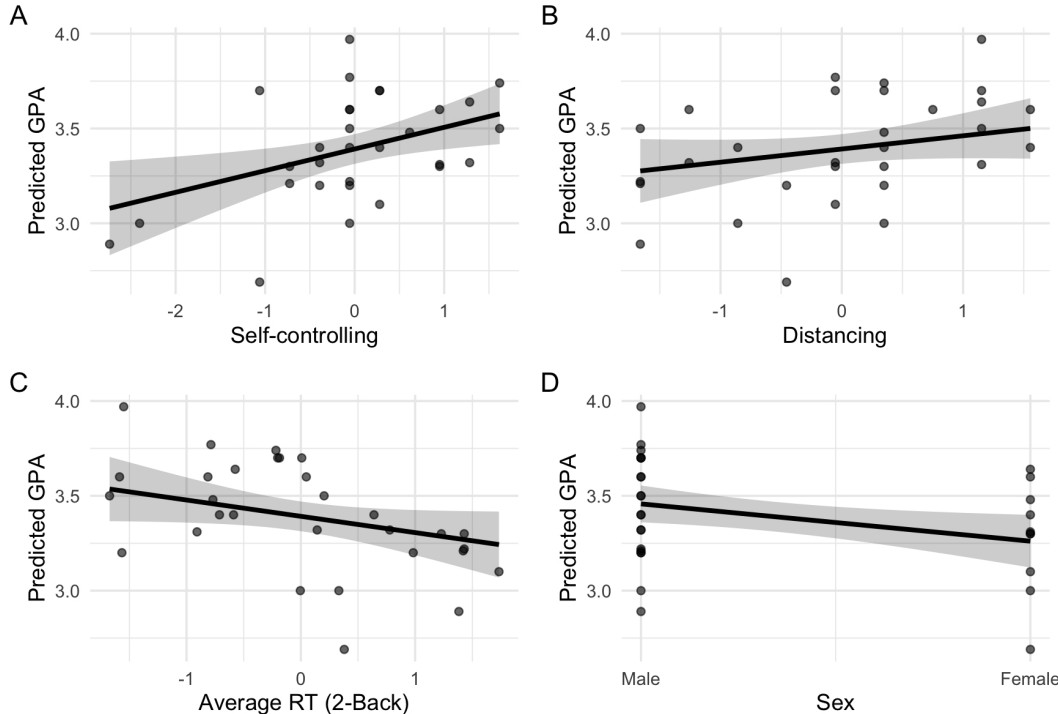

**Fig 3. Plotted graph regression model analysis description for associated variables.** Predicted effects of psycho-social and cognitive predictors on academic achievement (GPA), based on the standardized regression model. Predicted effects of psycho-social and cognitive predictors on academic achievement (GPA), based on the standardized regression model. Each panel displays one of the four predictors included in the final model: (A) self-controlling coping style, (B) distancing coping style, (C) average reaction time in the 2-back working memory task, and (D) sex. Solid lines represent the regression-predicted values of GPA, with shaded areas indicating 95% confidence intervals. Individual data points are overlaid to illustrate the distribution of observed values for each predictor.

GPA can be seen as the flip side of that observation – it reinforces the idea that students who can regulate their emotions and impulses under stress (instead of being overwhelmed by them) perform better academically. This is consistent with broader theories of self-regulation in education: for example, students with higher self-control and problem-focused coping tend to have superior academic outcomes, as they can stay task-focused and persevere through challenges. Our result is in line with studies that discovered that effective coping and self-management skills predict higher achievement in university students [3,42]. Conversely, the marginally negative impact of distancing coping fits with the literature on avoidant coping. While distancing oneself from problems can sometimes provide short-term emotional relief, it may impede active problem-solving and engagement with one's studies. Past research has generally found that avoidant or denial-based coping is associated with worse academic or adjustment outcomes, especially when prolonged. Our data hint at this trend – students who often tried to "tune out" academic stress had somewhat lower performance – echoing the notion that avoidance fails to address the root of academic difficulties (e.g., not studying for a tough exam) and can thus undermine achievement. Although this particular effect did not reach significance, it aligns with the theoretical expectation that proactive coping is more beneficial for students than disengaging from stress [3].

The observed gender difference (male students outperforming females) contrasts with some previous findings in Ethiopian higher education. In our prior study, female students have been reported to achieve equal or even better academic outcomes compared to males [3]. However, in the present sample, male students demonstrated higher GPAs, as it was also obtained in research by other authors [13–16].

Several explanations are possible. It may reflect differences in stress levels, academic engagement, or external responsibilities, such as family obligations, that disproportionately impact female students in this cohort. It is also possible that male students in this sample had greater access to academic resources or support networks. These findings highlight the importance of considering gender-specific challenges in educational interventions and call for further investigation into the mechanisms behind gender differences in academic achievement in low-resource settings.

The inclusion of cognitive predictors in this study allows us to compare their influence with that of psycho-social factors. We hypothesized that working memory and inhibitory control would play a role, based on substantial evidence from other populations [21,23]. Interestingly, our results did not show strong effects for ADHD symptoms. This somewhat contrasts with findings in high-income settings where cognitive abilities often emerge as significant predictors of academic success [42]. One possible reason is that among university students (especially those who have already navigated a challenging education system), there may be a restriction of range in cognitive abilities – i.e., most students in our sample might have at least adequate attention, since those with severe deficits may not have made it to higher education. Furthermore, some research suggests that the predictive power of basic executive functions on academic performance is stronger in earlier schooling years and may taper off at the university level [43]. By young adulthood, academic performance might depend more on domain-specific knowledge and study skills, with cognitive capacity differences playing a smaller role compared to in childhood. Our null finding for ADHD symptoms is somewhat surprising given the well-documented link between ADHD and academic problems [23]. It could be that the sample's self-reported ADHD scores were generally low or that students with significant ADHD-related impairments are underrepresented (possibly due to dropout or lack of diagnosis). It is also plausible that students with mild attention difficulties have learned compensatory strategies to cope in the university environment, thereby diluting the correlation between ADHD symptom scores and GPA. In summary, our results suggest that psycho-social factors (like coping and self-management) might overshadow cognitive factors in this particular context, or that the latter's effects are subtler and require larger samples or more sensitive measures to detect.

Overall, our findings partially supported the proposed hypotheses. Consistent with H1, psycho-social factors, particularly coping strategies and sex, were significant predictors of academic achievement. Specifically, higher use of self-controlling coping was associated with better GPA, and a significant gender difference in GPA was observed, although the direction of this effect (male students outperforming female students) differed from that found in our previous study.

Regarding H2, cognitive variables showed only limited associations with academic performance. Faster reaction time on the Go/No-Go task showed a trend toward predicting higher GPA, but ADHD symptoms did not significantly relate to GPA. Thus, while cognitive factors hinted at a potential influence, psycho-social factors appeared to have a stronger and more consistent impact in this sample.

### 4.3. Implications for low-resource educational settings

These findings carry several implications for understanding and improving academic achievement in low-resource contexts such as Ethiopia. First, the prominent role of coping strategies highlights that how students handle the pressures of university life can be just as important as traditional cognitive ability or prior academic preparation. In an environment where students may face numerous stressors – financial difficulties, large class sizes, limited instructional resources [3], and personal or family responsibilities – the capacity to self-regulate, stay organized, and manage stress is crucial. This suggests that interventions aimed at improving academic success should incorporate a focus on psycho-social skills. For example, universities might develop programs to train students in effective stress-coping techniques, time management, and emotional resilience. Such programs could help students maintain productivity and motivation even under challenging conditions. Second, our integration of cognitive assessments, albeit showing modest effects, implies that cognitive control is another piece of the puzzle. Even in a low-resource setting, students with slightly faster processing speed or better concentration tended to do better academically. This implies that identifying students with cognitive difficulties (e.g., very slow information processing or attentional lapses) and offering support could be beneficial. In practice, this might involve

screening for learning difficulties or attention deficits early in students' academic careers and providing targeted assistance (such as tutoring, cognitive training exercises, or counseling and medical support for those with ADHD). Moreover, the interplay between coping and cognition in our study underscores a holistic view: a student who is calm and self-controlled under stress is likely better able to leverage their cognitive resources for studying. Conversely, even a bright student may perform poorly if they cannot manage stress or procrastination. Thus, interventions that bridge both domains – for instance, study skills workshops that address both effective learning strategies (cognitive/meta-cognitive skills) and stress management (coping skills) – could be especially impactful in boosting GPA and reducing risk of academic failure.

### 4.4. Study limitations

While this study provides novel insights, several limitations must be acknowledged:

- The relatively small sample size and single institution can hinder statistical power and generalizability of the findings.
- The cross-sectional nature of study could not showed cause and effect relationship.
- The online data collection method was highly susceptible for response bias.
- The current study included only two working memory and inhibition with two specific paradigms (2-back and Go/No-Go) cognitive parameters which was not sufficient to summarize the cognitive predictors of students' academic success.
- The current study focused only on individual psycho-social and cognitive domains. However, the combined effects of contextual and environmental factors with cognitive domains on students' academic success in low-resource settings were not explored.

### 4.5. Future research

Building on this work, future studies should aim to address the above limitations and further explore the psycho-social–cognitive nexus in student achievement. Future research should be conducted as multi-center studies with larger sample sizes to enhance representativeness and the statistical power of predictors. Employing longitudinal, qualitative, and interventional designs would allow for a better understanding of causality and the real-world impact of different predictors in academic settings. It would also be valuable to include multiple cognitive domains to clarify the specific contribution of each parameter to academic outcomes. Additionally, the development of supportive environments that provide cognitive strategy training (e.g., memory aids, attention exercises) alongside counseling on stress management could be evaluated for their effects on GPA and student retention. Moreover, the present study provides a foundation for policymakers, higher education stakeholders, and peer support groups to design comprehensive strategies that address psycho-social, cognitive, environmental, and institutional factors influencing academic achievement in low-resource settings. Such approaches would deepen our understanding of how diverse factors interact to shape academic trajectories and ultimately help educators craft evidence-based strategies to strengthen student success.

## 5. Conclusion

In conclusion, this study contributes to the literature on academic achievement by integrating psycho-social and cognitive perspectives in the context of Ethiopian higher education. Extending the work of our prior systematic review and survey, we demonstrated that both domains offer valuable insights: on one hand, students' coping strategies and self-regulatory behaviors are strongly linked to their academic success, and on the other hand, cognitive processes like attention and working memory play a supportive role. Our findings underscore that academic achievement is not determined by intellect alone – it is equally shaped by how students manage stress, stay motivated, and deploy their cognitive resources in challenging learning environments. This integrated approach provides a more holistic understanding of student performance,

especially in a low-resource setting where students face multiple challenges. By examining psycho-social factors (such as coping mechanisms) alongside cognitive factors (such as executive functioning and ADHD symptoms), we bridged two traditionally separate research areas and highlighted their interaction in influencing GPA.

The insights gained from this study have practical implications for educational interventions. Universities and colleges – particularly in low-resource contexts – can use this evidence to design programs that support both the mindset and the skillset of students for optimal learning. For instance, institutions might consider the following applications:

- Incorporate workshops or courses that teach students effective coping strategies (e.g., problem-solving, time management, mindfulness techniques). By strengthening students' ability to handle academic stress and setbacks, such programs can enhance persistence and performance.

- Develop mentoring or counseling initiatives to improve students' academic self-perception and confidence. Positive reinforcement, goal-setting exercises, and role models may help students (especially those who doubt their abilities) to build self-efficacy, which in turn can motivate higher achievement.

- Implement screening for attention or learning difficulties early in the academic journey. Students who show signs of ADHD or executive function weaknesses could be offered targeted support services – for example, study skills coaching, cognitive strategy training, or referrals for medical evaluation if appropriate. Even simple adjustments like providing quiet study spaces or organizational tools can aid those with attentional challenges.

- Acknowledge gender patterns in academic performance and ensure that support services are accessible and tailored to all students.

Overall, this study underlines that improving academic outcomes in higher education calls for a comprehensive approach. Educational stakeholders should aim to cultivate both the cognitive capabilities and the psycho-social well-being of students. By integrating interventions that enhance executive functions (like working memory and attention control) with those that foster healthy coping and self-regulation, higher education institutions can create an environment in which a greater number of students thrive academically. Such multifaceted support is especially pivotal in resource-limited settings, where maximizing the efficacy of existing talents and mitigating stressors can make a substantial difference in student success. In sum, our research affirms that the synergy of strong psycho-social skills and cognitive skills forms a foundation for academic achievement, and leveraging this synergy through informed interventions holds promise for elevating educational attainment in Ethiopia and beyond.

## Supporting information

**S1 Data. Dataset.**
(XLSX)

**S1 File. Inclusivity in global research questionnaire.**
(DOCX)

## Acknowledgments

The author would to thank students who participated in the study

## Author contributions

**Conceptualization:** Chalachew Kassaw, Valeriia Demareva.

**Data curation:** Chalachew Kassaw, Valeriia Demareva.

**Formal analysis:** Chalachew Kassaw, Valeriia Demareva.

**Funding acquisition:** Chalachew Kassaw, Valeriia Demareva.

**Investigation:** Chalachew Kassaw, Valeriia Demareva.

**Methodology:** Chalachew Kassaw, Valeriia Demareva.

**Project administration:** Chalachew Kassaw, Valeriia Demareva.

**Resources:** Chalachew Kassaw, Valeriia Demareva.

**Software:** Chalachew Kassaw, Valeriia Demareva.

**Supervision:** Chalachew Kassaw, Valeriia Demareva.

**Validation:** Chalachew Kassaw, Valeriia Demareva.

**Visualization:** Chalachew Kassaw, Valeriia Demareva.

**Writing – original draft:** Chalachew Kassaw, Valeriia Demareva.

**Writing – review & editing:** Chalachew Kassaw, Valeriia Demareva.

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
