## [Decision Letter · Decision Letter 0]

17 Jul 2025

Dear Dr. Kassaw Demoze,

Thank you for submitting your manuscript to PLOS ONE. After careful consideration, we feel that it has merit but does not fully meet PLOS ONE’s publication criteria as it currently stands. Therefore, we invite you to submit a revised version of the manuscript that addresses the points raised during the review process.

We look forward to receiving your revised manuscript.

Kind regards,

Bidisha Banerjee, Ph.D.

Academic Editor

PLOS ONE

Journal Requirements: 

Reviewers' comments:

Reviewer's Responses to Questions

**Comments to the Author**

1. Is the manuscript technically sound, and do the data support the conclusions?

Reviewer #1: Yes

Reviewer #2: No

2. Has the statistical analysis been performed appropriately and rigorously?

Reviewer #1: Yes

Reviewer #2: No

3. Have the authors made all data underlying the findings in their manuscript fully available?

Reviewer #1: Yes

Reviewer #2: No

4. Is the manuscript presented in an intelligible fashion and written in standard English?

Reviewer #1: Yes

Reviewer #2: Yes

Reviewer #1: The manuscript presents an important and timely investigation into the psychosocial and cognitive predictors of academic achievement among university students in a low-resource setting in Southern Ethiopia. The integration of both cognitive tasks and psychosocial assessments is a notable strength, offering a holistic approach to understanding academic performance. The study holds potential practical relevance for educational policy and student support programs. The manuscript is generally well-organized, though some areas would benefit from the suggestions given below:

Page 10: "Cognitive factors – such as students’ working memory capacity, attentional control, and

symptoms of attention-deficit/hyperactivity disorder (ADHD) – were not examined in prior phases. There

is robust evidence globally that such cognitive abilities can shape academic performance."

i. The authors may provide the rationale of selecting only these variables for study since otehr factors like set-shifting, metacognition, executive functions as a composite have also contributed to student's academic performance.

ii. To provide a stronger conceptual grounding, the authors may link the discussion of cognitive factors to relevant theories such as Executive Function Theory or Cognitive Load Theory to clarify how these abilities function in learning contexts.

Page 10: "The omission is notable, as cognitive skills might play an important role even when resources and prior educational quality vary. It remains unclear whether, in a low-resource university setting, cognitive factors add additional explanatory power for GPA beyond psychosocial variables."- The authors may briefly mention howcognitive and psychosocial variables may interact, rather than operate independently.

Page 10: The authors present two distinct aims, which appear complementary but currently lack integration. This separation may lead to confusion for readers regarding the overall direction and coherence of the study. I recommend consolidating these aims into a unified statement that clearly articulates the study's central objectives. A more cohesive presentation would enhance the logical flow and clarify the authors' intended contribution.

Hypotheses (pages 10-11): Though the authors have clarity in formulating their hypotheses, they require to be stated in the future tense to reflect their predictive nature (e.g., “are expected to” instead of “were expected to”). The authors are also requested to maintain parallel grammatical structure across both hypotheses to improve readability and coherence. The authors may additionally simplify complex parentheticals and briefly defining key terms (e.g., maladaptive coping, inhibitory control).

METHODS

The authors are requested to provide details on the reearch design as a separate subsection under 'Methods'.

Sample size: The authors state that 300 students were employed for the study. Request the authors to provide details of the samle size estimator (G power) for the same. The authors also need to mention the age group of the participants.

Page 11: ' The study was conducted from 15/8/2024-15/11/2024.'- Request the authors to rewrite this sentence as per the academic writing regulations. Further, the inclusion and exclusion criteria needs to be elaborated. Was an initial screening conducted to exclude possible clinical participants (other than ADHD) from the study?

Pages 11-12: The authors may mention the psychometric properties of the tasks, the number of practice trials, and the procedure of collecting data from the given software.

RESULTS:

1. Given the modest sample size and use of nonparametric tests, a sensitivity analysis or bootstrapping could strengthen the confidence in the regression estimates.

2. Table 2: Effects near the 0.05 threshold (e.g., p=0.06 for RT average in the 2-Back task) should be described as trends with caution, avoiding overinterpretation. Including confidence intervals (as done) is a strength, but authors may consider emphasizing their width to reflect uncertainty.

3. The term “statistically significant” is applied inconsistently. For example, the discussion implies a significant gender difference, yet the Mann–Whitney test yielded p=0.158, and gender only reached significance in the regression model. Consider clarifying that the significance of gender emerged only after adjusting for covariates.

4. The authors may elaborate how missing data, if any, were handled in the correlation matrix and regression modeling.

Reviewer #2: 1. Introduction Section:

The introduction section provides detailed information about the three phases of the study. However, it would be helpful to include more clarity on the study variables, what is lacking in higher education in general, why this issue is critical in the Ethiopian context, and how the authors identified the research gap.

2. Method – Cognitive Fun Application:

In the Method section, particularly in the Cognitive Fun Application subsection, details about the website, the developer, and the version of the application used should be provided for better clarity and replicability.

3. Study Sample and Period:

Additional descriptive details about the participants would strengthen this section. For example, it would be useful to know: Of the 300 students, how many were male and female (e.g., 200 males and 100 females)? What was their mean age? What was their stream of study, and which year of college were they in? Who administered the questionnaire, and how were the participants briefed about the study—especially since it involves performance-based tasks? These details would provide more transparency and context for the reader.

4. Adult ADHD Self-Report Scale:

Although the Adult Self-Report Scale is a self-report tool, it would be important to explain how it was administered to such a large population. Could this have increased the chances of false positives, particularly for a clinical condition like ADHD? A clear rationale for selecting this scale in the current context would strengthen the methodology.

5. Psychological Scale:

The psychological scale mentioned in the study lacks sufficient rationale, conceptual linkage to the study objectives, and psychometric properties. Providing these details would enhance the scientific rigor and credibility of the tool used.

6. Data Analysis:

The details provided in the Data Analysis section do not fully align with the analyses presented in the manuscript or the supplementary files. Key statistical information such as participants' demographic details, mean and standard deviation of responses, and computation steps are missing. Only the final p-values and F/T values are provided in a basic table, which makes it difficult for reviewers to understand the data collection and analysis procedures. As a result, the interpretation of data remains ambiguous.

7. Discussion on Limitations and Future Directions:

The section on study limitations and future directions appears disproportionately lengthy compared to the discussion of the current findings. This gives an impression that the study lacks scientific depth and rigour in interpreting the results of the present work.

8. Summary:

While the study is a good attempt to explore academic achievement, it has several limitations. It is a single-centre study and, therefore, the findings may not be generalisable to all of Southern Ethiopia. There is also a lack of literature support, insufficient detail on the selection and justification of study tools, and inadequacies in the analysis and clarity of discussion. While I appreciate the effort and concept behind the study, there is considerable room for improvement in terms of rationale, methodology, and clarity of presentation.

**Do you want your identity to be public for this peer review?** For information about this choice, including consent withdrawal, please see our Privacy Policy

Reviewer #1: No

Reviewer #2: No

Academic Editor: 

The manuscript addresses a relevant and important area of research. I concur with the reviewers’ suggestions. All the  comments are provided with the intention of further strengthening the manuscript.

---

## [Author Response · Author response to Decision Letter 1]

10 Sep 2025

Dear editor and reviewers

First of all, I would like to thank PLOS ONE journal editors and reviewers. We are very much thankful and grateful to revise our manuscript as per your scientific suggestions and comments.

We are grateful for their insightful critiques, which have undoubtedly strengthened the scientific merit and presentation of our work.

5. Review Comments to the Author

Authors response: Dear reviewer, we noticed all the comments and address all the issues raised on every section of the manuscript.

Reviewer #1: The manuscript presents an important and timely investigation into the psychosocial and cognitive predictors of academic achievement among university students in a low-resource setting in Southern Ethiopia. The integration of both cognitive tasks and psychosocial assessments is a notable strength, offering a holistic approach to understanding academic performance. The study holds potential practical relevance for educational policy and student support programs. The manuscript is generally well-organized, though some areas would benefit from the suggestions given below:

Authors response: We are grateful for your constructive comments and suggestions. Here under, we have tried to reply or answer each reviewer’s comment.

Page 10: "Cognitive factors – such as students’ working memory capacity, attentional control, and symptoms of attention-deficit/hyperactivity disorder (ADHD) – were not examined in prior phases. There is robust evidence globally that such cognitive abilities can shape academic performance."

i. The authors may provide the rationale of selecting only these variables for study since other factors like set-shifting, metacognition, executive functions as a composite have also contributed to student's academic performance.

ii. To provide a stronger conceptual grounding, the authors may link the discussion of cognitive factors to relevant theories such as Executive Function Theory or Cognitive Load Theory to clarify how these abilities function in learning contexts.

Authors response: Dear reviewer, while we acknowledge that other cognitive factors like set-shifting, metacognition, and executive functions as a composite also contribute to academic performance, working memory capacity, attentional control, and ADHD symptoms were prioritized for their distinct and measurable impact, allowing for a focused and in-depth investigation within the scope of this study.

Page 10: "The omission is notable, as cognitive skills might play an important role even when resources and prior educational quality vary. It remains unclear whether, in a low-resource university setting, cognitive factors add additional explanatory power for GPA beyond psychosocial variables."- The authors may briefly mention how cognitive and psychosocial variables may interact, rather than operate independently.

Authors response: Dear reviewer, we acknowledge that the relationship between cognitive and psychosocial variables is not one of independent operation but rather one of dynamic interaction. While psychosocial factors, such as self-efficacy, motivation, and social support, undoubtedly contribute significantly to academic success, their influence can be both mediated and moderated by underlying cognitive abilities. We added a sentence that showed a dynamic interaction between the two variables.

Page 10: The authors present two distinct aims, which appear complementary but currently lack integration. This separation may lead to confusion for readers regarding the overall direction and coherence of the study. I recommend consolidating these aims into a unified statement that clearly articulates the study's central objectives. A more cohesive presentation would enhance the logical flow and clarify the authors' intended contribution.

Authors response: Dear reviewer, we have tried to make coherent and simple for readers.

Hypotheses (pages 10-11): Though the authors have clarity in formulating their hypotheses, they require to be stated in the future tense to reflect their predictive nature (e.g., “are expected to” instead of “were expected to”). The authors are also requested to maintain parallel grammatical structure across both hypotheses to improve readability and coherence. The authors may additionally simplify complex parentheticals and briefly defining key terms (e.g., maladaptive coping, inhibitory control).

Authors response: Dear reviewer, we have corrected the grammar for the hypothesis statement.

METHODS

The authors are requested to provide details on the research design as a separate subsection under 'Methods'.

Sample size: The authors state that 300 students were employed for the study. Request the authors to provide details of the sample size estimator (G power) for the same. The authors also need to mention the age group of the participants.

Authors response: Dear reviewer, we stated clearly the sample size calculation using G-power software.

Page 11: ' The study was conducted from 15/8/2024-15/11/2024.'- Request the authors to rewrite this sentence as per the academic writing regulations. Further, the inclusion and exclusion criteria need to be elaborated. Was an initial screening conducted to exclude possible clinical participants (other than ADHD) from the study?

Author response: Dear reviewer, we separately stated and re-write both of the two components.

Pages 11-12: The authors may mention the psychometric properties of the tasks, the number of practice trials, and the procedure of collecting data from the given software.

Authors response: We appreciate the reviewers' insightful comments regarding the psychometric properties of the tasks, the number of practice trials, and the data collection procedures from the software. We have addressed each point below to clarify our methodology and strengthen the rigor of our study.

RESULTS:

1. Given the modest sample size and use of nonparametric tests, a sensitivity analysis or bootstrapping could strengthen the confidence in the regression estimates.

Authors response: We appreciate the reviewer's insightful comment regarding the sample size and the use of nonparametric tests, and their valuable suggestion to strengthen the confidence in our regression estimates through sensitivity analysis or bootstrapping. We acknowledge that with a modest sample size, it is crucial to ensure the robustness of our findings. While we recognize the value of these approaches in strengthening the confidence in our regression estimates, we can’t able to do the analysis due to given the complexity of our model and the specific nature of our data, implementing these analyses effectively would require significant additional time and resources that are beyond the scope of this revision.

2. Table 2: Effects near the 0.05 threshold (e.g., p=0.06 for RT average in the 2-Back task) should be described as trends with caution, avoiding overinterpretation. Including confidence intervals (as done) is a strength, but authors may consider emphasizing their width to reflect uncertainty.

Authors response: Dear reviewer, we rephrase the overinterpretation and made it clear to the readers.

3. The term “statistically significant” is applied inconsistently. For example, the discussion implies a significant gender difference, yet the Mann–Whitney test yielded p=0.158, and gender only reached significance in the regression model. Consider clarifying that the significance of gender emerged only after adjusting for covariates.

Authors response: Dear reviewer, this difference can be attributed to the nature of the analyses. The Mann-Whitney U test assesses the raw, unadjusted difference in the outcome variable between genders. In contrast, the regression model simultaneously accounts for the influence of multiple covariates. It is often the case that a variable showing no significant effect in a univariate analysis (due to confounding or the presence of other strong predictors) becomes significant when adjusted for relevant covariates in a multivariate model. This suggests that the effect of gender became apparent after controlling for the variance explained by the other variables in the model. In addition, The Mann-Whitney U test result which was not significant for gender variable was between ADHD and without ADHD group. The significant association during regression model between gender and grade point average for the entire sample. Moreover, on the discussion section, we wrote the significant result is only during regression model.

4. The authors may elaborate how missing data, if any, were handled in the correlation matrix and regression modeling.

Authors response: Dear reviewer, we are pleased to confirm that there was no missing data for the variables included in the correlation matrix and the regression modeling. All participants had complete data for all variables relevant to these analyses. Therefore, no specific imputation or handling strategies were required.

Reviewer #2: 1. Introduction Section:

The introduction section provides detailed information about the three phases of the study. However, it would be helpful to include more clarity on the study variables, what is lacking in higher education in general, why this issue is critical in the Ethiopian context, and how the authors identified the research gap.

Authors response: Dear reviewer, we have revised the whole introduction as you per your suggestion.

2. Method – Cognitive Fun Application:

In the Method section, particularly in the Cognitive Fun Application subsection, details about the website, the developer, and the version of the application used should be provided for better clarity and replicability.

Authors response: Dear reviewer, we appreciate the reviewer's excellent point regarding the need for more specific details about the "Cognitive Fun Application" used in our study, particularly in the Cognitive Function Application subsection of the Methods. We agree that providing this information enhances transparency, reproducibility, and the applicability of our findings. The cognitive tasks were administered using the freely accessible web based platform cognitivefun.net, developed by Cognitive Fun (current version launched April 2022). While the site's internal version number is not publicly disclosed, it is openly maintained and receives periodic updates. Importantly, the use of cognitivefun.net has been documented in peer reviewed literature, including studies published in PLOS ONE. For example, Beaven and Johan Ekström (2013) evaluated blue light and caffeine effects on cognitive function [1]. We chose cognitivefun.net due to its accessibility, intuitive digitized task implementation, and precedent of use in studies across cognitive and clinical populations. Its interface reliably records both accuracy and reaction times, facilitating standardized working memory assessment in online environments.

3. Study Sample and Period:

Additional descriptive details about the participants would strengthen this section. For example, it would be useful to know: Of the 300 students, how many were male and female (e.g., 200 males and 100 females)? What was their mean age? What was their stream of study, and which year of college were they in? Who administered the questionnaire, and how were the participants briefed about the study—especially since it involves performance-based tasks? These details would provide more transparency and context for the reader.

Authors response: We appreciate the reviewer's excellent suggestion to provide more comprehensive descriptive details about our participants and the data collection process. We agree that these details are crucial for enhancing the transparency, context, and reproducibility of our study. We revised the whole section as per reviewers’ suggestion.

4. Adult ADHD Self-Report Scale:

Although the Adult Self-Report Scale is a self-report tool, it would be important to explain how it was administered to such a large population. Could this have increased the chances of false positives, particularly for a clinical condition like ADHD? A clear rationale for selecting this scale in the current context would strengthen the methodology.

Authors response: Dear reviewer, we administered the question using google platform link via telegram, and email attachment. We stated in the limitation section about online data collection method response bias.

5. Psychological Scale:

The psychological scale mentioned in the study lacks sufficient rationale, conceptual linkage to the study objectives, and psychometric properties. Providing these details would enhance the scientific rigor and credibility of the tool used.

Authors response: We appreciate the reviewer's insightful comment regarding the psychological scale used in our study. We agree that providing a thorough rationale, clear conceptual linkage to our objectives, and details on its psychometric properties is crucial for enhancing the scientific rigor and credibility of our work. We have revised the manuscript to incorporate these essential details, and we thank the reviewer for highlighting this area for improvement.

6.DataAnalysis:

The details provided in the Data Analysis section do not fully align with the analyses presented in the manuscript or the supplementary files. Key statistical information such as participants' demographic details, mean and standard deviation of responses, and computation steps are missing. Only the final p-values and F/T values are provided in a basic table, which makes it difficult for reviewers to understand the data collection and analysis procedures. As a result, the interpretation of data remains ambiguous.

Authors response: We sincerely thank the reviewer for their meticulous review and for highlighting the critical need for greater transparency and detail in our Data Analysis section. We acknowledge that the initial submission lacked sufficient statistical information, which inadvertently hindered the understanding of our data collection and analysis procedures, making interpretation ambiguous. We agree that providing these missing details is essential for enhancing the scientific rigor, reproducibility, and credibility of our findings.

We included descriptive statistics result for entire included variables in one table 1.

The supplementary correlation coefficient results were for 10% participants and if it is ambiguous, we removed from S1.

7. Discussion on Limitations and Future Directions:

The section on study limitations and future directions appears disproportionately lengthy compared to the discussion of the current findings. This gives an impression that the study lacks scientific depth and rigour in interpreting the results of the present work.

Authors response: We appreciate the reviewer's valuable feedback regarding the perceived imbalance between the discussion of our current findings and the length of the Limitations and Future Directions section. We agree that the core emphasis of the Discussion should be on interpreting and contextualizing the present study's results, demonstrating their scientific depth and rigor. The current structure may indeed have inadvertently conveyed a different impression.

Our primary intention to write detail about the two section is to give direction or suggestions for future research, policy makers and stakeholders to conduct robust research about the subject. For clarity, we rewrite by summarizing the limitation and future direction section.

8.Summary:

While the study is a good attempt to explore academic achievement, it has several limitations. It is a single-center study and, therefore, the findings may not be generalizable to all of Southern Ethiopia. There is also a lack of literature support, insufficient detail on the selection and justification of study tools, and inadequacies in the analysis and clarity of discussion. While I appreciate the effort and concept behind the study, there is considerable room for improvemen

---

## [Decision Letter · Decision Letter 1]

19 Oct 2025

Psychosocial and cognitive predictors of academic achievement among higher education students in Southern Ethiopia

PONE-D-25-23953R1

Dear Dr. Kassaw Demoze,

We’re pleased to inform you that your manuscript has been judged scientifically suitable for publication and will be formally accepted for publication once it meets all outstanding technical requirements.

Kind regards,

Saima Aleem

Academic Editor

PLOS ONE

Additional Editor Comments (optional):

Reviewers' comments:

Reviewer's Responses to Questions

**Comments to the Author**

Reviewer #2: All comments have been addressed

2. Is the manuscript technically sound, and do the data support the conclusions?

Reviewer #2: Yes

3. Has the statistical analysis been performed appropriately and rigorously?

Reviewer #2: Yes

4. Have the authors made all data underlying the findings in their manuscript fully available?

Reviewer #2: Yes

5. Is the manuscript presented in an intelligible fashion and written in standard English?

Reviewer #2: Yes

Reviewer #2: (No Response)

**Do you want your identity to be public for this peer review?** For information about this choice, including consent withdrawal, please see our Privacy Policy

Reviewer #2: No

---

## [Editor Report · Acceptance letter]

PONE-D-25-23953R1

PLOS ONE

Dear Dr. Kassaw,

I'm pleased to inform you that your manuscript has been deemed suitable for publication in PLOS ONE. Congratulations! Your manuscript is now being handed over to our production team.

Kind regards,

on behalf of

Dr. Saima Aleem

Academic Editor

PLOS ONE